# Effects of controversial contexts on opinion changes through discussion and evaluation of decisions: A group decision experiment regarding the issue of removed soil in Fukushima

Yume Souma[1,2☉*], Yukihide Shibata[1‡], Mie Tsujimoto[1‡], Qinglin Cui[1‡], Takashi Nakazawa[3☉], Tomoyuki Tatsumi[4☉], Yoshiko Arima[5☉], Susumu Ohnuma[1,6☉]

**1** Department of Behavioral Science, Faculty of Humanities and Human Sciences, Hokkaido University, Sapporo, Hokkaido, Japan, **2** Research fellow of the Japan Society for the Promotion of Science, Tokyo, Japan, **3** Department of Sociology, Toyo University, Tokyo, Japan, **4** Department of Life and Career Planning, Toyohashi Sozo Junior College, Toyohashi, Aichi, Japan, **5** Department of Psychology, Kyoto University of Advanced Science, Kyoto, Japan, **6** Center for Experimental Research in Social Sciences, Hokkaido University, Sapporo, Hokkaido, Japan

☉ These authors contributed equally to this work.
‡ YS, MT, and QC also contributed equally to this work.
* souma.yume.l6@elms.hokudai.ac.jp

## Abstract

Designing settings that enable structured deliberation is crucial, particularly for contentious public issues, as debates can sometimes shift opinions toward extremes, create social division, and make it difficult to evaluate decisions positively. Constructive controversy has been proposed to avoid such breakdowns. However, only a few empirical studies have been conducted in the context of contentious public issues. This study aims to examine how competitive and cooperative controversies influence opinion change and decision evaluation in the context of public decision-making by conducting a group decision experiment. The discussion topic was the final disposal of the removed soil outside Fukushima Prefecture, which was caused by a nuclear power plant accident. The removed soil is currently stored in facilities in the towns of Okuma and Futaba, Fukushima Prefecture. By law, the government must complete the final disposal of the removed soil outside Fukushima Prefecture by 2045. The participants were 128 Japanese university students. This study manipulated controversial contexts as independent variables. One was a competitive context condition, whereby participants refuted opposing opinions, and the other was a cooperative context condition, whereby participants contemplated both pros and cons thoroughly. Each group consisted of four participants, and there were 32 groups in total (16 per condition). Participants in both conditions discussed and decided whether to approve the final disposal of the removed soil outside Fukushima Prefecture. The results indicated no significant differences in opinion changes between the conditions. However, participants in the cooperative condition evaluated their decisions more positively

**Data availability statement:** All relevant data for this study are publicly available from the OSF repository (https://osf.io/f8e62).

**Funding:** This work was supported by the Environment Research and Technology Development Fund (https://www.env.go.jp/policy/kenkyu/suishin/english/gaiyou/index.html) (Grant Number [JPMEERF22S20907] and [JPMEERF20251001] to S.O.); the Japan Science and Technology Agency (https://www.jst.go.jp/EN/) (Grant Number [JPMJCR20D1] to S.O.); the Japan Society for the Promotion of Science (https://www.jsps.go.jp/english/) (Grant Number [23K22343] to S.O.; Grant Number [24KJ0296] to Ym.So.); and the Graduate Grant Program of the Graduate School of Humanities and Human Sciences, Hokkaido University (https://www.let.hokudai.ac.jp/general/review-fee-support) (Grant Number [2] to Ym.So.). These funding agencies played no role in the study design, data collection and analysis, the decision to publish, or the preparation of the manuscript.

**Competing interests:** The authors have declared that no competing interests exist.

than those in the competitive condition did. Notably, under competitive conditions, participants whose pre-discussion opinions were not reflected in the groups' decisions tended to rate outcomes less favorably. These findings contribute to a deliberative design and lead to more structured discussions on public issues.

## Introduction

The literature on group decisions has noted that group discussions can shift individual opinions, as demonstrated by laboratory experiments [1,2] and case studies [3]. However, a change in opinion is not necessarily desirable or undesirable in itself. What matters is whether the shift emerges through thoughtful deliberation or merely reflects conformity to the majority or to dominant voices, such as groupthink [4,5]. This distinction in the process is crucial because it determines the quality of the group decision [6,7] and the satisfaction of the group members [8].

This issue becomes particularly salient in the context of public decision-making. In recent years, scholars and policymakers have emphasized the importance of deliberative democracy [9,10], which provides opportunities for citizens with diverse perspectives to engage in debates about policies related to urban planning, environmental management, and other public concerns. Constructive citizen deliberation can strengthen the legitimacy of collective decisions and lead participants to accept resulting policies. Moreover, citizen involvement in policymaking can improve the quality of decisions [11]. At the same time, when citizens fail to deliberate in a structured way, especially on contentious issues, public discussions risk deepening social divisions.

Therefore, it is crucial to design settings that enable structured deliberation on public issues. One prominent example of such design is Deliberative Polling [3,12,13]. Deliberative Polling involves conducting an initial survey with a randomly selected sample, providing balanced information materials to those who agree to participate, and then convening an event in which participants engage in small-group discussions and address questions to experts in plenary sessions. The researchers conducted a follow-up survey after deliberations. Sunstein [14] argued that Deliberative Polling tends to polarize group opinions. Conversely, Curato et al. [15] emphasized that participants often shifted toward more moderate positions, suggesting that polarization does not occur during deliberations. Opinion changes in deliberative arenas, such as Deliberative Polling, do not necessarily emerge in every form of group discussion. When the conditions for deliberation are properly established, group opinions tend to move in a moderate direction rather than become more extreme, as shown by case studies [16,17] and laboratory experiments [18,19]. Specific conditions for deliberation include the establishment of common ground before the discussion [20], the provision of balanced and impartial factual information [3,18], the presence of deliberative rules or moderators [19], and exposure to heterogeneous perspectives [21,22]. Satisfying these conditions plays a critical role in preventing polarization.

Researchers have proposed and implemented deliberation designs, such as Deliberative Polling, to promote thoughtful discussion, and empirical studies have identified the conditions that prevent opinion polarization. However, previous research on deliberation has paid relatively little attention to how the structure of discussion and interaction processes among participants affect collective outcomes. This study focuses on these points by providing empirical evidence from laboratory experiments on group decision-making.

The structure–process–outcome theory [23] offers a framework for analyzing the relationship between a discussion and its outcomes. According to this theory, situational structures determine the interaction process, which, in turn, determines the outcome. From this perspective, the design of deliberation itself can be understood as part of situational structures. How do structures influence the discussions among participants with various opinions?

A helpful approach for integrating divergent opinions is constructive controversy ([24], among others). Constructive controversy is defined as "when one person's ideas, information, conclusions, theories, and opinions are incompatible with those of another, and the two seek to reach an agreement" [24]. Previous studies have shown that constructive controversy generates higher-quality solutions than debate or concurrence seeking [25,26] and stimulates the production of more ideas [27]. Johnson and Johnson [28] provides a comprehensive review.

However, merely engaging in controversy with individuals who hold different views does not always yield high-quality outcomes. One critical point for making a controversy constructive is the presence of a cooperative goal structure. Deutsch [29] argued that when people face conflict situations, the conflict resolution process may follow either a competitive or cooperative process, and this distinction leads to different forms of interaction and outcomes. Competitive processes involve negative goal interdependence, where the extent to which one person achieves their goals reduces the extent to which others can achieve theirs. Cooperative processes involve positive goal interdependence, in which individuals' goal attainment mutually supports each other. Based on Deutsch [29], controversy inherently entails some form of conflict, it can become truly constructive only when resolution follows a cooperative process.

## Effects of constructive controversy

Constructive controversy also plays a critical role in civil-political discourse [30]. However, research on public participation, including Deliberative Polling, has rarely examined how the context of controversy influences the two key outcomes of public deliberation–opinion change and the evaluation of group decisions. This study addresses this gap by focusing on two aspects of opinion change, group polarization and opinion homogenization, and examining their implications for how participants evaluate group decisions.

Deliberation raises concerns about group polarization [1,2,14] and opinion homogenization [31]. Group polarization refers to the phenomenon observed in social psychology on group decision-making, where "members of a deliberating group (to) move to a more extreme position, with the direction of the shift determined by the majority or average of the members' pre-deliberation preferences" ([32], p. 334). Opinion homogenization refers to the reduction in opinion variance within a group through deliberation, resulting in greater uniformity of views among members [33].

Given evidence from Deliberative Polling, when participants deliberate on diverse perspectives of issues, the group's average opinion often shifts in a more moderate direction [34]. Although individuals inherently tend to process information in ways that reinforce their pre-existing views [35,36], exposure to diverse perspectives helps mitigate biases, strengthening decision-making [37]. Therefore, some scholars argue that proper deliberative processes prevent polarization [15]. Luskin et al. [31] analyzed data from 21 Deliberative Polling projects and found that opinion shifts toward moderation occurred more frequently than shifts toward the extremes. However, this discussion does not guarantee that polarization will not occur. For example, laboratory experiments by Grönlund et al. [18] and Strandberg et al. [19] demonstrated that the presence of deliberative norms, rather than the composition of group opinions (homogeneous versus heterogeneous), plays a more decisive role in determining whether polarization emerges.

Political discussions that shift opinion toward extremes along ideological or partisan lines can generate political and emotional divisions within society, which are already pressing problems [38–42]. At the same time, the opposite outcome—convergence of opinions toward a uniform stance after deliberations—also raises concerns since it suggests non-deliberative dynamics at work [22]. Luskin et al. [31] examined opinion homogenization and found a modest but consistent tendency toward homogenization in Deliberative Polling.

Although research on deliberation often treats polarization and homogenization as problematic, this study does not necessarily consider them negative. Indeed, some deliberative theorists argue that homogenization resulting from compromise among participants may represent a desirable outcome [43]. Likewise, suppose that participants reach a consensus, accept decisions, and voluntarily shift their opinions without coercion. In this case, such a change in opinion, although it may appear to be the same as polarization or homogenization, can be interpreted positively. Conversely, even if the group's average opinion and variance remain the same, the discussion may still qualify as deliberative if individuals shift in different directions after considering multiple aspects of the issue.

Thus, the mere occurrence of an opinion change does not, in itself, determine whether an outcome is positive or negative. A critical aspect that has been frequently overlooked in constructive controversy research is the manner in which participants evaluate group decisions. While some studies have examined attitudes toward decision-making, they typically assessed decision quality, such as creativity [28,44] and commitment to a task [45], rather than participants' evaluations of whether the decision itself is good. In the context of public participation, decision evaluation is a central component of outcome evaluation and strongly related to decision acceptance [46,47]. Hence, a decision evaluation is indispensable when considering the design of public decision-making discussions.

How, then, does the context of controversy—competitive versus cooperative—influence opinion change and decision evaluation? A competitive context fosters closed-mindedness, leading participants to reject opposing ideas and information [30]. It creates situations where winners and losers inevitably emerge based on who offers the "best" ideas or knowledge [48]. In this context, the participants pursue mutually incompatible goals and resolve conflicts through competitive processes. For instance, in a debate over a dichotomous choice where only one of two competing options can be adopted, such as economic growth vs environmental preservation, the goal supporting one side is ultimately incompatible with the other side's goal. In such situations, even though considering the opposing viewpoint provides new advantages or disadvantages of one's preferred option, doing so is often difficult and may lead people to remain entrenched in suboptimal solutions.

In contrast, a cooperative context fosters open-mindedness, encouraging participants to listen to opposing views, gain a more accurate understanding of those views, and move toward positions that integrate their own perspectives with those of others [49,50]. It also considers differences in perspectives and knowledge as valuable resources to be combined into a well-understood decision [48]. In this setting, participants work together to achieve shared objectives and progress toward conflict resolution through cooperative processes. An example of the situations is that discussants present their best proposals from their respective perspectives toward a shared goal, such as achieving the SDGs, encompassing both economic growth and environmental preservation, and strive to refine them through constructive mutual exchange toward integration.

Regarding opinion change, competitive controversies tend to leave opposing views in conflict, making it unlikely that the group's average opinion will shift toward either moderation or extremes. In contrast, cooperative controversies encourage participants to listen and seek mutual understanding, which is more likely to shift the group's mean opinion—either toward moderation or toward extremes. Furthermore, competitive controversies discourage compromise; therefore, the variance in opinions within a group is unlikely to converge. By contrast, cooperative controversies promote the search for common ground that could facilitate convergence in opinion variance.

In decision evaluation, competitive controversies discourage participants from understanding opposing views and often lead to outcomes that reflect only one side. Participants whose perspectives were not incorporated into the group decision

tend to evaluate it less favorably, thereby lowering the group's average decision evaluation. In contrast, cooperative controversies encourage participants to consider opposing views and integrate them into a comprehensive decision, thereby increasing satisfaction across participants and yielding a higher overall decision evaluation.

## Discussion topic

This section describes the subject of the group decision experiment. This study addressed the issue of disposing of removed soil outside of Fukushima Prefecture.

The Fukushima Daiichi nuclear power plant accident in March 2011 resulted in the release of large amounts of radioactive substances into the environment, causing environmental pollution. The Japanese government conducted decontamination work to reconstruct the damaged areas, which produced vast volumes of soil and other materials. The government initially placed the removed soil in temporary storage sites in a broad area of Fukushima Prefecture. However, to proceed with the restoration and reconstruction of Fukushima, a facility was necessary to safely and centrally manage and store the removed soil. Consequently, an interim storage facility was constructed in the towns of Okuma and Futaba in Fukushima Prefecture. The Fukushima Governor accepted the facility on the condition that the final disposal of the removed soil would be conducted outside Fukushima Prefecture.

The area of the interim storage facility is approximately 16 km², of which about 80% is privately owned by villagers [51]. Before the accident, many villagers lived there and used their land for paddy fields and shrines. The amount of removed soil and other materials delivered to the facility is approximately 14 million meter ³. Of the massive amount of removed soil, approximately 25% was high-concentration soil exceeding 8,000 Bq/kg, and 75% was low-concentration soil below 8,000 Bq/kg [51].

Regarding the decontaminated soil, the government must take necessary measures to complete final disposal outside Fukushima Prefecture within 30 years from the start of transportation to the interim storage facility (by March 2045), in line with the Japan Environmental Storage & Safety Corporation Law (Law No. 44 of 2003, promulgated on May 16, 2003). The government enacted the law because Fukushima residents, including those in the towns of Okuma and Futaba, were already overburdened. However, the specific procedure and disposal site for the final disposal have not yet been determined.

From the perspective of obtaining a site area, it is challenging to dispose of the large amounts of removed soil outside Fukushima Prefecture. Therefore, the key to realizing it is to reduce the final disposal volume [51]. To ensure safety, the government is attempting to convert low-concentration soil below 8,000 Bq/kg into recycled material and promote its reuse, provided that the management body and responsibility are guaranteed [52]. The evidence for 8,000 Bq/kg or less for recycling does not exceed 1 mSv/year, which is the lower value of the reference level for radiation protection strategies recommended [53]. Low-concentration soil is usable for public utilities such as road construction, levees, bridge girders, and farmland development in raised earth foundations. Demonstration projects for soil reuse were conducted in Minami-Soma City and Iitate Village, Fukushima Prefecture, and safety was confirmed in both projects [51,54,55].

The Japanese government has emphasized the need for public understanding and debate on the removed soil [51]. Similarly, the IAEA [56] recommends broad public deliberations involving diverse citizens. One of the central issues for such deliberation is whether the promotion of final disposal outside Fukushima Prefecture, as mandated by law, should be approved [57–63]. According to a survey, awareness of this legal requirement remains quite low in Japan except for Fukushima [64]. Low awareness of the law implies a lack of understanding of its legitimacy, namely that Fukushima residents have already borne significant burdens, and that the prefectural governor accepted the interim storage facility on the condition that the final disposal would be completed outside Fukushima Prefecture. Unless the public recognizes and accepts both the meaning of the law and the necessity of final disposal outside Fukushima Prefecture, the law will not be practically effective. Thus, before selecting a site for final disposal, early and ongoing communication and engagement regarding the final disposal of the removed soil is important to create robust deliberative processes and strengthen decision-making on the issue.

When discussing the final disposal of removed soil, the reuse of low-concentration soil also should be addressed because promoting soil reuse reduces the total volume of soil requiring final disposal. Regarding soil reuse, previous group decision experiments have confirmed that participants' opinions tend to shift positively through discussion [60,65]. In contrast, opinions on final disposal outside Fukushima Prefecture can shift either positively or negatively depending on the group's decision [60]. Therefore, this study focuses on the deliberations involved in deciding whether to implement final disposal outside Fukushima Prefecture. However, since soil reuse and final disposal are inseparable, we establish a setting in which participants can decide on final disposal while discussing both topics.

Although conflicts over this issue have not yet been revealed, it is possible that opinions on implementing final disposal outside Fukushima Prefecture are divided. Consequently, administrators should pay careful attention to the design of deliberative forums. Therefore, the issue provides an appropriate context for comparing competitive and cooperative controversies in public decision-making.

Applying the constructive controversy theory to the topic of the removed soil, the following predictions are delivered. In a competitive controversy, participants are likely to adhere firmly to their initial positions, whether in favor of or against the final disposal outside Fukushima Prefecture. Consequently, the group's mean opinion is unlikely to shift, and opinion variance is unlikely to decrease. However, in a cooperative controversy, participants are more inclined to listen to opposing perspectives, making opinion shifts and reductions in variance more likely. Moreover, when groups reach a decision on the final disposal, competitive controversies will tend to foster rigidity, prevent mutual understanding, and result in a decision that reflects only one aspect of the argument. Therefore, the decision evaluation could remain relatively low. Conversely, cooperative controversies promote the careful consideration of opposing views and increase the likelihood of integrated outcomes, leading to relatively higher evaluations of group decisions.

## Research aim and hypotheses

This study empirically examines how competitive and cooperative controversies influence opinion change and decision evaluation in the context of public decision-making. To this end, a group decision experiment was conducted, manipulating the context of the controversy. Given earlier predictions in the context of public deliberation on the final disposal of removed soil outside Fukushima Prefecture, this study tested the following hypotheses.

H1-a. Compared with competitive controversies, cooperative controversies led to greater shifts in the group's average opinion on the final disposal outside Fukushima Prefecture, in either a moderate or an extreme direction.

H1-b. Compared with competitive controversies, cooperative controversies result in greater reductions in opinion variance within groups regarding the final disposal outside Fukushima Prefecture.

H2. Compared with competitive controversies, cooperative controversies yield relatively higher evaluations of group decisions regarding the final disposal outside Fukushima Prefecture.

## Methods

### Experimental overview

This study was approved by the Ethics Review Committee of Center for Experimental Research in Social Sciences, Hokkaido University (Approval Code: R4-05, approved on April 21, 2023). The experiment was conducted between June and October 2023. The participants discussed and decided whether the final disposal of the removed soil outside Fukushima Prefecture was acceptable. The experiment employed a one-factor, two-level, between-subjects design. Controversial contexts were manipulated. One was the competitive condition, characterized by clearly different positions in which participants were instructed to refute the opponent, and the other was the cooperative condition, in which participants were asked to consider both positions regardless of their initial opinion. The details are provided below.

## Participants

All participants were Japanese university students, and discussions were conducted in groups of four. The group size was set at four to reduce the likelihood of decisions being made solely by majority votes and ensure that every member had sufficient opportunity to exchange opinions within a limited time. Participants were recruited via the Sona System, an online platform, from June 12, 2023, to October 2, 2023. The Sona System has over 2,000 registered users. Registration was open to any individual who wished to participate as an experimental subject, with no restrictions on faculty affiliation or academic year. In this system, experimenters post available timeslots for the study, and registered participants can sign up for the slots that best fit their schedules. Participants were not informed of the experiment's detailed content in advance; they received a full explanation upon arrival at the laboratory. All participants were previously unacquainted with each other. A flat payment of 1,500 yen (approx. 10 USD) was given to all participants as a token of appreciation, regardless of the decision.

Considering the statistical power and moderate effect size in the t-test (statistical power $\beta = 0.80$, effect size Cohen's [66] $d = 0.50$, $p = .05$), a sample size of 16 groups, with 64 participants, was determined for each condition. This sample size is set assuming individual-level analysis, hence it would be insufficient when conducting group-level analysis. The final sample consisted of 93 first-year students (competition condition: 50; cooperation condition: 43), 16 second-year students (competition: 5; cooperation: 11), 10 third-year students (competition: 3; cooperation: 7), and nine fourth-year students (competition: 6; cooperation: 3). Regarding academic disciplines, 38 participants were from the humanities and social sciences (competition: 19; cooperation: 19), and 90 from sciences and engineering (competition: 45; cooperation: 45). In terms of gender, 70 participants were identified as male (competition: 38; cooperation: 32), 57 as female (competition: 25; cooperation: 32), and one as neither male nor female/preferred not to answer (competition: 1; cooperation: 0). These demographic variables were not controlled when the groups were formed. The high proportion of first-year students is likely due to their increased motivation to participate in psychological experiments immediately after registering for the Sona System. Additionally, the predominance of students from science and engineering disciplines reflects the university's overall student demographics. Although there were imbalances in faculty affiliation and academic year, the demographic composition was consistent between conditions. Therefore, this did not pose a problem for comparing the results between conditions.

Before the discussion, participants completed a pre-questionnaire to gather information about their backgrounds. None of the participants reported being from the Fukushima Prefecture. They also answered questions about their prior knowledge and interest in the issue of removed soil, and their responses are summarized in Tables 1 and 2.

The level of interest in the issue was comparable between the two conditions, with many participants reporting little to no interest. Most participants responded to the knowledge items by indicating that they "did not know about any of them."

Assignment to the experimental conditions was based on participants' initial position on the final disposal outside Fukushima Prefecture, which was measured in the pre-questionnaire (two options: support or oppose). After reviewing the responses, the experimenters assigned groups to conditions such that the initial distribution of opinions within each group was as balanced as possible. Table 3 presents the initial opinion compositions for each condition.

## Creation of informed materials

Before conducting the experiment, the researchers prepared information material to explain the issue of removed soil to participants. To develop this material, a consultation was held with experts from multiple fields, including radiation medicine, soil contamination, and risk communication, who evaluated the accuracy, neutrality, and appropriateness of the content. Specifically, they were asked to assess whether the information was balanced and whether participants who were unfamiliar with the issue could reasonably understand it. The researchers revised the material repeatedly until all experts considered it satisfactory.

**Table 1. Prior knowledge regarding the disposal of removed soil.**

| Before participating in the experiment, did you already know the following? | Competitive cond. | Cooperative cond. |
|---|---|---|
| | n (%) | n (%) |
| The final disposal of removed soil outside Fukushima Prefecture is stipulated by law. | 4 (6.3%) | 2 (3.1%) |
| Removed soil is being stored temporarily in Okuma and Futaba Towns, Fukushima Prefecture. | 8 (12.5%) | 12 (18.6%) |
| The majority of the removed soil is low-concentration material at 8,000 Bq/kg or below. | 1 (1.6%) | 3 (4.7%) |
| Did not know any of them. | 54 (84.4%) | 50 (78.1%) |

The denominator for n was 64 under each condition. The values in the table represent the number of people who selected each option and the percentage within the conditions. Except for "Did not know any of them," participants could select multiple options.

**Table 2. Prior interest in the disposal of removed soil.**

| Interest in the disposal of removed soil | Competitive cond. | Cooperative cond. |
|---|---|---|
| | n (%) | n (%) |
| I had absolutely no interest. | 13 (20.3%) | 14 (21.9%) |
| I didn't have much interest. | 26 (40.6%) | 23 (35.9%) |
| It's hard to say either way. | 10 (15.6%) | 8 (12.5%) |
| I had somewhat of an interest. | 14 (21.9%) | 18 (28.1%) |
| I had a strong interest. | 1 (1.6%) | 1 (1.6%) |

The values in the table represent the number of people who selected each option and the percentage within the conditions. Participants could select only one option.

**Table 3. The number of initial opinion compositions.**

| Number of initial opinions for/against the final disposal of removed soil outside Fukushima Prefecture in each group | | Number of applicable groups | |
|---|---|---|---|
| Pros (n) | Cons (n) | Competitive cond. | Cooperative cond. |
| 4 | 0 | 5 | 3 |
| 3 | 1 | 6 | 7 |
| 2 | 2 | 3 | 4 |
| 1 | 3 | 1 | 2 |
| 0 | 4 | 1 | 0 |
| total | | 16 | 16 |

The final information material consisted of 19 slides. It included an overview of the process from the generation of the removed soil to its final disposal, as well as statements made by the mayors of Okuma and Futaba, where the interim storage facility is located, explaining the circumstances under which they accepted the facility. Moreover, the information materials included explanations of soil reuse, the distinction between soil reuse and final disposal, and that soil reuse reduces the volume of soil required for final disposal. The participants had access to the material at all times during the discussions.

In the pre-questionnaire, the participants were asked to evaluate the neutrality, difficulty, and length of the informational material. The results are summarized in Table 4. Responses to items one to five were recorded on a seven-point Likert

**Table 4. Evaluation of informational materials.**

| questionnaire items | | Competitive cond. | | Cooperative cond. | |
|---|---|---|---|---|---|
| | | Mean | SD | Mean | SD |
| 1 | The information provided in the informational materials is neutral.[a] | 5.28 | 1.53 | 5.22 | 1.64 |
| 2 | The information provided in the informational materials is well-balanced. [a] | 5.28 | 1.35 | 5.12 | 1.51 |
| 3 | The information provided in the informational materials is biased. [a] | 2.52 | 1.37 | 2.75 | 1.57 |
| 4 | The information provided in the informational materials is designed to arbitrarily lead the discussion.[a] | 2.84 | 1.71 | 2.84 | 1.65 |
| 5 | The information provided in the informational materials is biased toward a particular position.[a] | 2.56 | 1.41 | 2.73 | 1.62 |
| 6 | The amount of information provided in the informational materials is insufficient/sufficient.[b] | 3.78 | 0.79 | 3.59 | 1.00 |
| 7 | The difficulty level of the information provided in the informational materials is easy/difficult.[c] | 4.03 | 0.85 | 4.23 | 0.85 |

[a] a seven-point Likert scale ranging from 1 ("disagree") to 7 ("agree").

[b] a seven-point Likert scale ranging from 1 ("too short") to 7 ("too much").

[c] a seven-point Likert scale ranging from 1 ("too easy") to 7 ("too difficult").

scale ranging from 1 ("disagree") to 7 ("agree"). Responses on item six were recorded on a seven-point Likert scale ranging from 1 ("too short") to 7 ("too much"), with four representing "just right." Responses on item seven were recorded on a seven-point Likert scale ranging from 1 ("too easy") to 7 ("too difficult"), with four representing "just right." The participants generally considered the material to be appropriate.

Informed materials are provided in the S1 Appendix.

## Experimental procedure

Upon arrival in the experimental room, participants first received the informed consent and the other consent form for audio and video recording, and they signed them if agreed. Participants were informed that they could withdraw from the experiment at any time if they did not agree to participate or chose to participate without audio and video recording. Next, the participants received the informational materials and watched a ten-minute instructional video covering the same content as the materials. Then, participants completed a pre-questionnaire. All the items included in the pre-questionnaire and their responses are presented in the S1 Table.

Subsequently, a discussion context of the assigned conditions is introduced for both conditions.

Afterward, participants received "individual opinion sheets," on which they wrote their ideas for individual deliberation before discussion, and were given five minutes to organize their opinions regarding the final disposal of removed soil outside Fukushima Prefecture and its reuse. If necessary, they were given a few additional minutes. Subsequently, the researcher reintroduced the discussion section. Participants were then instructed to record the group's decision on a "sheet" by the end of the discussion. Each group was given only one sheet. Participants were required to select one option from the sheet. Participants were also informed that, while a group decision on soil reuse was not mandatory, it could be a key point when discussing the merits of final disposal outside Fukushima Prefecture. Additionally, the researcher said that one member of each group would present their group's decision, the reasons behind it, and the process leading to the decision after the discussion. The experimenter did not determine who was present; instead, group members selected the presenter voluntarily.

The discussion lasted 40 minutes. No facilitator was present; instead, participants managed and facilitated the discussion themselves, as this study intended to observe interactions solely among participants and to ensure the effects of controversial contexts. After the discussion, one participant from each group presented the group's decisions, with the others able to provide additional comments, if necessary. Finally, participants completed a post-questionnaire survey. All items included in the post-questionnaire and the responses are provided in the S2 Table.

## Manipulation

To ensure that the participants adopted the competitive or cooperative controversy cognitive set, the following manipulations were implemented, as shown in Table 5.

After the experimenter instructed participants in the discussion methods, participants answered questions to confirm their understanding. Specifically, they were asked, "Regarding the upcoming discussion, please select one of the following options that best describes what you should do." The two options were: (1) "Assert your own position so that it becomes the group's decision, refute or persuade others with differing opinions, and decide whether to approve or disapprove the final disposal of removed soil outside Fukushima Prefecture," or (2) "Consider both supporting and opposing viewpoints, examine both the advantages and disadvantages of the final disposal of removed soil outside Fukushima Prefecture, and decide collectively what should be done on the issue." In the competitive condition, 22 out of 64 participants selected option 1, and 42 selected option 2. In the cooperative condition, two out of 64 participants selected option 1, and 62 selected option 2.

## Analysis

To verify the effectiveness of the controversial contexts manipulation, participants rated the extent to which they engaged in these discussion methods in the post-questionnaire: "asserting my position and persuading opponents with different opinions, so that your position would be adopted as the group decision" or "considering both supporting and opposing perspectives and deliberating the merits and demerits of the final disposal outside Fukushima Prefecture." Responses were recorded on a six-point Likert scale ranging from 1 ("did not try to engage") to 6 ("tried to engage").

Table 5. Manipulation in the contexts of controversy.

| | Competitive cond. | Cooperative cond. |
|---|---|---|
| position plate: visual markers to publicly display each participant's initial opinion (support or opposition) based on their pre-questionnaire responses. | placed in front of each participant | Not placed |
| discussion methods | Participants were instructed to<br>• assert your position and persuade opponents with different opinions, so that your position will be adopted as the group decision<br>• decide whether to approve or disapprove the final disposal outside Fukushima Prefecture by the end of the discussion | Participants were instructed to<br>• consider both supportive and opposing stances on the final disposal of the removed soil, critically sift through the merits and demerits of the final disposal outside Fukushima Prefecture<br>• decide what should be done by the end of the discussion |
| individual opinion sheet: sheets given to each participant for writing down their individual views before discussions | Participants were asked to fill in<br>1. their position on the final disposal outside Fukushima Prefecture (as written on the "position plate")<br>2. arguments supporting their position<br>3. assumed arguments from the opposing position<br>4. arguments to counter the assumed opponents' arguments | Participants were asked to fill in<br>1. their position on the final disposal outside Fukushima Prefecture (the same position they answered in the pre-discussion questionnaire)<br>2. the merits of the final disposal outside Fukushima Prefecture<br>3. the demerits of the final disposal outside Fukushima Prefecture<br>4. agreeable ideas to varied individuals with various stances, considering both supportive and opposing contentions |
| decision entry sheet: sheets given to each group for filling in their group decision | Options:<br>"Approve," "Disapprove," or "Not decided" for the final disposal outside Fukushima Prefecture | Options:<br>"Approve," "Disapprove," "Other" (entry specified contents), or "Not decided" for the final disposal outside Fukushima Prefecture |

To test H1-a and H1-b, i.e., opinion change, participants were requested to rate their position on "conducting the final disposal of removed soil outside Fukushima Prefecture" in both the pre- and post-questionnaires. Responses were recorded on a six-point Likert scale ranging from 1 ("disapprove") to 6 ("approve").

Following Luskin et al. [31], a polarization index was calculated to examine whether the group's mean opinion shifted in a more extreme or moderate direction after the discussion. The degree of polarization $P_g$ was calculated using Equation (1), where $\overline{A_{gt}}$ represents the mean opinion of Group $g_{th}$ at time-$t$. In this study, $t=1$ refers to the pre-discussion phase, and $t=2$ to the post-discussion phase.

$$P_g = \left(\overline{A_{g2}} - \overline{A_{g1}}\right) S_{g1} \tag{1}$$

In this index, $P_g > 0$ indicates polarization and $P_g < 0$ indicates moderation. Because the response scale ranged from 1 to 6 in this study, $S_{g1}$ was defined that $S_{g1} = 1$ when the pre-discussion group mean $\overline{A_{g1}}$ exceeded 3.5, and $S_{g1} = -1$ when $\overline{A_{g1}}$ was below 3.5. In other words, $S_{g1}$ took a value of 1 if the pre-discussion group's mean opinion was more favorable than the midpoint, and −1 if it was less favorable. As an exception, when $\overline{A_{g1}} = 3.5$, the group was regarded as polarized regardless of whether $\overline{A_{g2}}$ was higher or lower than 3.5, and $P_g$ was calculated as $P_g = \left|\overline{A_{g2}} - \overline{A_{g1}}\right|$. Using the $P_g$ obtained, H1-a was tested.

A homogenization index following Luskin et al. [31] was calculated to examine whether within-group opinion variance converged or increased after the discussion. The degree of homogenization $H_g$ was calculated using Equation (2), where $s_{gt}$ denotes the standard deviation of group $g_{th}$ at time-$t$. In this study, $t=1$ refers to the pre-discussion phase, and $t=2$ to the post-discussion phase.

$$H_g = s_{g1} - s_{g2} \tag{2}$$

In this index, $H_g > 0$ indicates homogenization, whereas $H_g < 0$ indicates variation. Using the $H_g$ obtained in this manner, H1-b was tested.

To test H2 regarding the evaluation of group decisions, two items in the post-questionnaire were included: "Overall, I think the group decision was good" and "All things considered, I can evaluate the group decision as good." Responses were given on a seven-point Likert scale ranging from 1 (strongly disagree) to 7 (strongly agree). The two items comprised the "evaluation of decision" scale (Cronbach's α = 0.97).

All analyses were performed using R software (version 4.4.1). The code used for the analyses is available in the repository (https://osf.io/f8e62/overview). The complete results of the regression analysis are shown in the S3 Table.

## Results

### Manipulation check

The manipulations were checked using the responses to a post-discussion questionnaire. A multilevel linear regression analysis was conducted with responses to each item as the dependent variable and the condition (0: competitive condition, 1: cooperative condition) as the predictor variable. Discussion groups were treated as random intercepts. For the item "asserting my position and persuading opponents with different opinions to be adopted as the group decision," a significant effect of condition was found ($\beta = -1.24$, 95% CI = [−1.85, −0.67]). Participants in the competitive condition reported making greater efforts to refute and persuade than did those in the cooperative condition ($M=4.03$, SD = 1.50; $M=2.78$, SD = 1.43, respectively). In contrast, for the item "considering both supporting and opposing perspectives and deliberating the merits and demerits of the final disposal outside Fukushima Prefecture," no effect of condition was found ($\beta = 0.26$, 95% CI = [−0.14, 0.65]). The responses were similar in the competitive (M = 4.98, SD = 1.13) and cooperative (M = 5.25, SD = 0.91) conditions.

## Group decisions

The distribution of initial opinions on the final disposal of the removed soil outside Fukushima Prefecture before discussions to compare the conditions was as follows: 45 supporters and 19 opponents in the competitive condition, and 43 supporters and 21 opponents in the cooperative condition, indicating a roughly equal distribution between the conditions. Table 6 presents the initial opinion compositions (number of supporters and opponents) and group decisions regarding the final disposal outside Fukushima Prefecture.

Notably, five of the 16 groups in the competitive condition did not reach a decision by the end of the discussion period, whereas all groups in the cooperative condition did. Instead, in the cooperative condition, six of the 16 groups selected "Other," which neither approved nor disapproved the final disposal outside Fukushima Prefecture, thus compromising both sides. For example, some groups concluded that the burden of soil disposal should be shared within the entire Tohoku region, or that disposal should take place in both uninhabited areas outside and within Fukushima Prefecture.

**Table 6. The initial opinion compositions and group decisions of each group.**

| Competitive condition | | | | | Cooperative condition | | | | |
|---|---|---|---|---|---|---|---|---|---|
| group id | initial opinion (pros-cons) | group decisions on the final disposal outside Fukushima | $P_g$ | $H_g$ | group id | initial opinion (pros-cons) | group decisions on the final disposal outside Fukushima | $P_g$ | $H_g$ |
| competitive 1 | 2–2 | Disapproval | −1.50 | −0.10 | cooperative 1 | 4–0 | Approval | 1.00 | 0.76 |
| competitive 2 | 3–1 | Not-decide | 0.00 | 0.00 | cooperative 2 | 3–1 | Other | 1.25 | 0.44 |
| competitive 3 | 2–2 | Approval | −2.00 | 0.00 | cooperative 3 | 3–1 | Approval | −1.75 | 0.68 |
| competitive 4 | 1-3 | Disapproval | −0.25 | 0.12 | cooperative 4 | 2–2 | Other | −1.00 | −0.82 |
| competitive 5 | 4–0 | Approval | 0.25 | 0.44 | cooperative 5 | 4–0 | Approval | 1.00 | 1.00 |
| competitive 6 | 3–1 | Approval | 1.25 | 0.38 | cooperative 6 | 3–1 | Approval | 2.00 | 1.63 |
| competitive 7 | 3–1 | Not-decide | 1.00 | 0.54 | cooperative 7 | 2–2 | Approval | 1.50 | 0.56 |
| competitive 8 | 0-4 | Disapproval | −3.25 | 1.32 | cooperative 8 | 3–1 | Other | −1.50 | 0.00 |
| competitive 9 | 4–0 | Approval | 1.50 | 0.24 | cooperative 9 | 2–2 | Other | 1.00 | 0.46 |
| competitive 10 | 3–1 | Not-decide | 0.25 | −0.66 | cooperative 10 | 3–1 | Approval | 1.50 | 0.18 |
| competitive 11 | 4–0 | Approval | 0.75 | 0.50 | cooperative 11 | 1-3 | Approval | −2.00 | 0.34 |
| competitive 12 | 3–1 | Not-decide | −0.50 | −0.32 | cooperative 12 | 4–0 | Approval | 1.25 | 0.79 |
| competitive 13 | 4–0 | Approval | −1.00 | −0.94 | cooperative 13 | 1-3 | Disapproval | −0.75 | −0.74 |
| competitive 14 | 3–1 | Not-decide | −1.00 | −0.42 | cooperative 14 | 3–1 | Disapproval | 1.50 | 1.25 |
| competitive 15 | 2–2 | Approval | 1.25 | 0.79 | cooperative 15 | 3–1 | Other | −0.75 | 0.50 |
| competitive 16 | 4–0 | Approval | 0.50 | 0.24 | cooperative 16 | 2–2 | Other | 1.50 | 0.58 |
| Ave. | | | −0.17 | 0.13 | | | | 0.36 | 0.48 |

## Opinion change

To examine H1-a and H1-b, analyses of the changes in participants' opinions regarding the final disposal of the removed soil outside Fukushima Prefecture were conducted. Fig 1 illustrates how group mean opinions and within-group variances changed from pre- to post-discussion for each group.

Using the formulas described in the Analysis section, two indices were calculated: $P_g$, which indicates whether the group mean opinion shifted toward a more extreme or moderate direction after discussions, and $H_g$, which indicates whether within-group opinion variance converged or increased after discussions. To test H1-a and H1-b, a two-way analysis of variance (2 [condition] × 3 [group decision]) on $P_g$ and $H_g$ was employed after applying an aligned rank transform [67], which enabled conducting an ANOVA for nonparametric data. In the competitive condition, there were three levels of group decision: "Approve," "Disapprove," and "Not-decide," while four levels in the cooperative condition: "Other," in addition to the above three. However, since no group reached the "Not-decide" option in the cooperative condition, the group decision in the cooperative condition was treated as three levels in the following analysis. Furthermore, to align levels between the competitive and cooperative conditions, the "Not-decide" in the competitive condition and the "Other" in the cooperative condition were treated as the same level for comparison. The unit of analysis was the group, with n = 16 for each condition.

The results of two-way ANOVA on $P_g$ showed no significant effects of the condition, group decisions, or their interaction ($F$ (1, 26) = 1.68, $p$ = .21, $\eta_p^2$ = 0.06; $F$ (2, 26) = 2.36, $p$ = .11, $\eta_p^2$ = 0.15; $F$ (2, 26) = 0.61, $p$ = .55, $\eta_p^2$ = 0.04, respectively). Thus, the data did not support H1-a, which predicted that the cooperative condition would produce greater shifts in group-mean opinions than would the competitive condition. Additionally, the overall mean of $P_g$ was 0.09 (95% CI = [−0.39, 0.58]), indicating that $P_g$ aggregating all groups was approximately zero.

The results of two-way ANOVA on $H_g$ revealed no main effect of condition or interaction between condition and group decisions ($F$ (1, 26) = 1.91, $p$ = .18, $\eta_p^2$ = 0.07; $F$ (2, 26) = 0.16, $p$ = .85, $\eta_p^2$ = 0.01, respectively). Hence, the data did not

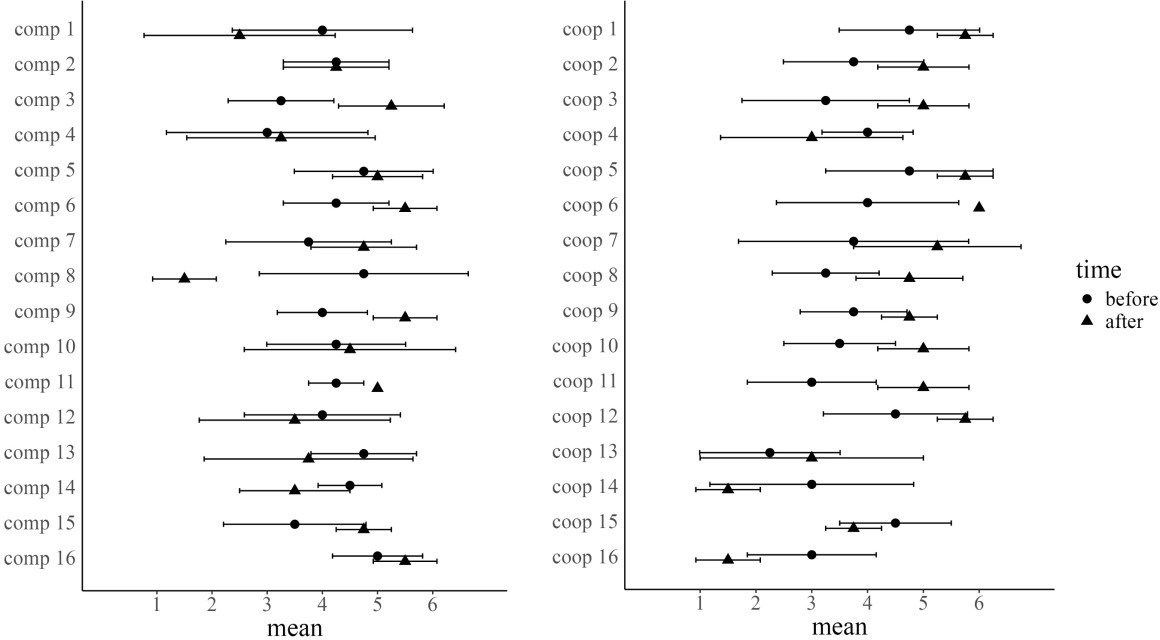

**Fig 1. Changes in group opinion mean and within-group variance (left side: competitive condition, right side: cooperative condition).** Error bar represents ±1SD.

support H1-b, which predicted greater reductions in the group-within-variance of opinions in the cooperative condition than in the competitive condition. However, group decisions had a moderate main effect ($F_{(2, 26)}$ = 2.64, $p$ = .09, $\eta_p^2$ = 0.16). Post hoc comparisons between decision levels were conducted, with $p$-values adjusted using the Bonferroni correction. The result suggested that groups classified as "Not-decide/Other" had lower ($M$ = 0.03, SD = 0.51) than groups that decided "approve" ($M$ = 0.48, SD = 0.54) to the final disposal outside Fukushima Prefecture ($EMMs$ = 8.40, $se$ = 3.66, $t$ ratio (26) = 2.29, $p$ = .09). This pattern suggests that groups categorized as "Not-decide/Other" exhibited less convergence in opinion variance. Moreover, groups that reached the "Not-decide" option in the competitive condition showed lower ($M$ = −0.17, SD = 0.46) than groups that reached the "Other" option in the cooperative condition ($M$ = 0.19, SD = 0.53). In other words, "Not-decide" groups under the competitive condition tended to display greater variance in opinions after discussion than before. Additionally, the overall mean of $H_g$ was 0.30 (95% CI = [0.08, 0.52]), indicating that $H_g$ aggregating all groups exceeded zero. This result suggests that, as a whole, opinion variances within groups tend to converge through discussions.

### Evaluation of group decisions

A multilevel linear regression analysis was conducted with participants' responses to the items assessing their evaluation of the group decision as the dependent variable and the condition (0: competitive, 1: cooperative) as the predictor variable. Discussion groups were treated as random intercepts. The analysis results revealed a significant main effect of condition on the evaluation of the group decision ($\beta$ = 0.85, 95% CI = [0.26, 1.40]), indicating that participants in the cooperative condition ($M$ = 5.59, SD = 1.23) rated the group decision more positively than those in the competitive condition ($M$ = 4.75, SD = 1.43). This finding supports H2, which predicts higher evaluations of decisions in the cooperative than in the competitive condition. To explore the patterns of responses in more detail, the means of the group decision evaluation for each combination of participants' initial opinions and the group decision were calculated, as shown in Fig 2.

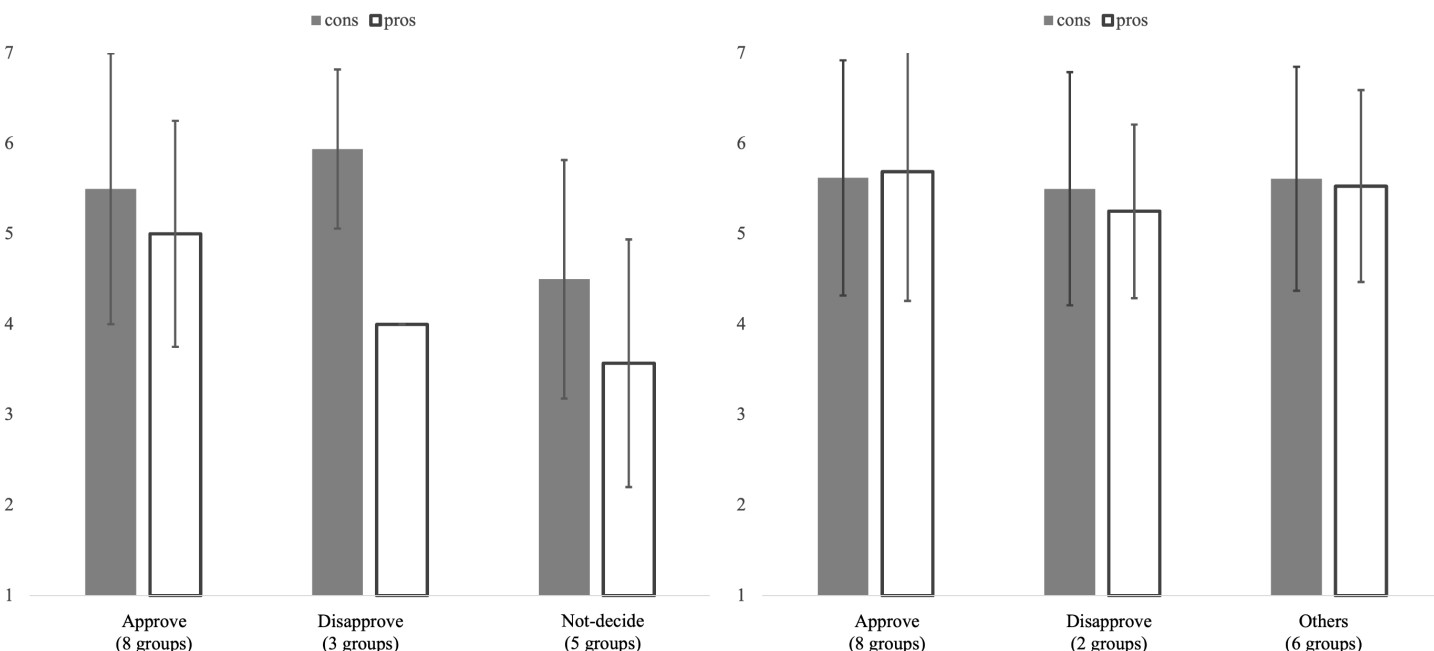

**Fig 2. Evaluation of the decisions by each category (left side: competitive condition, right side: cooperative condition).** Error bar represents ±1SD.

Among the group that decided to disapprove of the final disposal of the removed soil outside Fukushima Prefecture under the competitive condition, all participants who initially favored the final disposal outside Fukushima Prefecture rated it as 4, resulting in an SD of 0.

In the competitive condition, participants whose initial opinions supported final disposal outside Fukushima Prefecture rated the group decision relatively lower when the group disapproved. Additionally, when the group did not reach a decision, participants tended to evaluate it less favorably, regardless of their initial opinion. In contrast, in the cooperative condition, participants generally rated the group decision relatively higher, regardless of whether it reflected their initial opinion, even when they selected the "Other" option.

## Discussion

### Summary of results

This study conducted a group discussion experiment that manipulated the context of controversy to empirically examine how competitive and cooperative controversies influence opinion changes and decision evaluations in public decision-making.

The results showed no evidence that groups in the cooperative condition shifted their mean opinions more than those in the competitive condition. Thus, H1-a was not supported. In fact, opinion changes under both conditions were subtle, moving neither clearly in an extreme nor in a more moderate direction. Similarly, the variance in opinions did not decrease more in the cooperative than in the competitive condition, while opinions tended to converge overall; therefore, H1-b was not supported. However, the analysis revealed differences depending on group decisions. Groups that selected the "Not-decide/Other" option displayed significantly greater increases in variance after discussion compared to groups that decided "approve" to final disposal outside Fukushima Prefecture. Moreover, only the groups in the competitive condition that did not reach a consensus ("Not-decide") showed a clear tendency for opinion variance to increase after discussion.

Regarding evaluations of group decisions, participants in the cooperative condition rated their group's conclusions more favorably than those in the competitive condition, supporting H2. Furthermore, within the competitive condition, evaluations tended to be particularly low among participants whose initial opinions did not align with the group's decision and those in groups that failed to reach a decision ("not-decide"). By contrast, participants in the cooperative condition rated their group's decisions as relatively high overall, even when their initial opinions did not align with the decision.

### Interpretation and implication of results

Unlike previous research indicating that participants' attitudes tend to change in constructive controversy in a laboratory experiment [68], the analyses revealed no significant differences between the competitive and cooperative conditions in terms of changes in group mean opinion and opinion variance. This result can be interpreted through the lens of motivated reasoning. Motivated reasoning generally refers to how people's goals or motivations affect their reasoning and judgments [69–72]; it is a psychological process in which individuals process information in ways that support their pre-existing beliefs. This concept has gained attention in the context of risk management and risk assessment [73] and is considered relevant to opinion formation and opinion change through deliberation. According to motivated reasoning, confirmation bias is expected to operate during opinion change, where individuals actively process information or arguments that align with their initial positions. Consequently, individuals should incorporate information that supports their pre-existing views, leading to more extreme opinions. In the present experiment, while opinions changed within each group, the average opinions did not necessarily become more extreme across all groups. If information processing had functioned solely to support initial opinions, it should have acted only to reinforce or polarize pre-existing views, which was not the case in our results. Therefore, the mutual influence between individual opinions and group dynamics warrant further detailed analysis.

Furthermore, the results of opinion change should be interpreted in relation to decision evaluation. In contrast to the competitive condition, participants in the cooperative condition evaluated their group's decisions favorably, even when

they were not aligned with their initial views. This result suggests that participants in the cooperative condition had accepted it sufficiently to evaluate the group's decisions positively, despite their opinion shifts not differing from those in the competitive condition. The higher level of acceptance in the cooperative condition likely arose from the possibility of formulating decisions that integrated multiple perspectives—illustrated by the six groups that reached "Other" decisions.

In summary, although the competitive and cooperative controversy contexts did not produce significant differences in opinion changes in group means and variances, evaluations of those decisions indicated differences. Specifically, opinion changes under the competitive condition tended to be irrelevant to the acceptance of group decisions, whereas those under the cooperative condition corresponded to it.

To interpret opinions on soil reuse alongside the decisions on final disposal, an additional analysis of the questionnaires was conducted. The results revealed a common trend across both conditions: participants held positive views of soil reuse before the discussion, and these evaluations became even more positive afterward (see S4 Table for detailed results). This finding is consistent with previous studies [58,60,65]. This suggests that by providing information and key points that allow discussion of soil reuse in conjunction with final disposal, participants in both conditions could deliberate from a broader perspective that extends beyond the simple binary of whether to implement final disposal outside Fukushima Prefecture.

The statements provided for the group decisions serve as a clue to cement the validity of this interpretation. Among the six groups in the cooperative condition who selected the "Other" option, two included the implementation of soil reuse in their group decision. Furthermore, one of these two groups explicitly stated that soil reuse should be promoted at multiple locations nationwide. Similarly, in the competitive condition, although the "Other" option was unavailable, multiple groups stated they had reached a consensus to proceed with soil reuse when explaining their group decisions. For example, one group whose decision was "Not-decided" agreed that soil reuse should be implemented, including in areas outside Fukushima Prefecture. These results suggest that because soil reuse was readily acceptable in both conditions, the overall conflict did not escalate or lead to significant polarization, even when opinions on the final disposal decision remained divided.

## Limitations

Before discussing the contributions of this study, two limitations should be noted. First, the cognitive setting of the competitive conditions was insufficient. In the competitive condition, an attempt was made to foster a more antagonistic context by placing "position plates" in front of each participant that presented their position (pros or cons) to make their positions and divergences explicit. Nonetheless, the post-questionnaire revealed that participants in the competitive condition tried to engage in discussions that considered both for- and against-positions to a similar extent as those in the cooperative condition. One possible reason for this is that the participants did not hold firm opinions on the removed soil issue. The pre-discussion questionnaire revealed that many had little prior knowledge or interest in the topic. Opinion formation regarding final disposal was probably fostered by participants' personal experiences, general beliefs (e.g., whether to prioritize efficient risk processing), and intuitive emotions (e.g., anxiety or dread about the removed soil, or emotional empathy for the Fukushima residents). However, it is also possible that the opinions did not reach the level of firm convictions, potentially due to a lack of knowledge or due to the inherent limitations of the information that could be provided in the materials before the discussions. Without unwavering views, even in a competitive context, participants may find it difficult to refute or challenge opposing opinions. This restriction may prevent competitive contexts from fully functioning. However, some evidence, such as the existence of groups that failed to reach a decision, supports the idea that participants perceive discussions as competitive. Considering these constraints and outcomes, the controversial contexts implemented in this study represent the most realistic and best possible attempt under these circumstances.

Moreover, when people with conflicting interests engage in direct debates, discussions can easily become contentious, leading to deeper conflicts and probably preventing consensus. This prediction highlights the need for caution when

designing deliberative processes. As a further step, while this study did not involve stakeholders such as residents of potential disposal sites for the removed soil, future research should conduct group discussion experiments—for example, "What if your community was selected as a disposal site of the removed soil?"—to examine the decision among stakeholders.

Second, the participants were university students residing in urban areas, and nobody was from Fukushima Prefecture. Accordingly, there has been little direct interest in removed soil. This may explain why participants, even in competitive conditions, could discuss calmly. Therefore, whether stakeholders with a direct interest—for example, residents of candidate disposal sites—would discuss in the same way requires careful consideration. At present, however, it is impossible to involve such stakeholders because the regions bearing the burden of the removed soil have not yet been determined. Nevertheless, for long-term issues such as the disposal of removed soil, the younger generation's engagement and involvement in decision-making, particularly in urban areas, is crucial. In this regard, this study provides valuable insights before actual public deliberations take place. However, perceptions of the issue and the sense of personal involvement likely vary across generations and regions. Therefore, further social experiments and public participation workshops involving a more diverse range of demographic attributes are required.

## Contributions and future directions

### Insights into the removed soil issue

This study provides insights into how controversial contexts influence group decisions made in public deliberations on the issue of removed soil. As in the competitive condition, when the choice options were dichotomous (e.g., approval or disapproval) and the controversy was framed to prompt participants to refute opposing views, only one side of the positions tended to be incorporated into the decisions. Consequently, some participants may have felt that the group's decisions did not reflect their views, making it difficult to evaluate them favorably. Such a dynamic implies that under circumstances with multiple stakeholders, the group decision may incorporate the claims of only particular stakeholder groups while disregarding others. For example, suppose the group decides to oppose final disposal outside Fukushima Prefecture. In this case, the views of those who will bear the burden of the removed soil are represented, but the concerns of residents in Okuma and Futaba towns, who hope for the final disposal outside Fukushima Prefecture, remain excluded from the decision.

In contrast, when the choice options included a third option beyond "for" or "against"—such as "Other," as in the cooperative condition—the groups could make a decision that integrated a broader range of participants' views. This result partly aligns with previous research indicating that controversies in cooperative contexts are more likely to lead to decisions that integrate opposing views [49]. In this study, six of 16 groups in the cooperative condition chose "Other," and participants in the condition consistently evaluated their decisions more positively. These findings suggest that cooperative controversy contexts can yield compromised solutions that reflect the perspectives of stakeholders on both sides, such as those in Okuma, Futaba, and potential disposal districts, as well as decisions that can be accepted even by those who originally opposed them; even more reach *aufheben*.

However, it should be noted that the decision options differed between the two conditions. Due to the absence of the "Other" option, some groups in the competitive condition might have led to undecided responses. On the other hand, the presence of the "Other" option per se in the cooperative condition does not guarantee reaching a compromise, and even with this option, it is theoretically possible not to reach an agreement. Nevertheless, the results revealed that compromised decisions occurred in several groups under the cooperative condition, suggesting that cooperative contexts can yield acceptable outcomes. Further examination verifying this is warranted.

The law regarding the final disposal of removed soil requires that it be conducted outside Fukushima Prefecture. Public understanding and opportunities for deliberation are vital to the law's effectiveness. Although conflicts over this issue

 

have not yet emerged in society, implementing the final disposal outside Fukushima Prefecture is likely to divide opinions, underscoring the importance of carefully designing deliberative processes.

Looking at the current situation, while dialogues with the Fukushima residents—who have already borne the burden of interim storage facilities—have been conducted intensively (e.g., [74,75]), there are few examples of symposia or workshops targeting the public (e.g., [58]). As noted in the IAEA recommendations [56], it is necessary to advance both public communication and stakeholder engagement. This study focused on public deliberations and provided valuable insights for designing and implementing them regarding the issue of removed soil, by experimentally demonstrating how controversial contexts shape group decisions. Future research should also examine the design of deliberations among stakeholders where conflict is more likely to arise, as well as the overall process design for public-stakeholder engagement that extends beyond specific participation opportunities or workshops.

### Suggestions for public deliberation

In addition to the issue of decontaminated soil, this study has broader implications for public deliberation. Incorporating the perspective of constructive controversy, an approach largely absent from previous research on deliberative design, this study examined how different contexts of controversy influence opinion changes and the evaluation of the group decisions. The results showed no significant differences between competitive and cooperative contexts in terms of opinion change alone. However, evaluations of group decisions varied by condition. Taken together, these findings suggest that, even when groups display similar patterns of opinion change, the extent to which participants accept the group decision may differ. Specifically, discussion itself affects individuals' views, though being both extreme and moderate are possible [33], yet in competitive contexts, such opinion change tends to be irrelevant to acceptance of the group decision. By contrast, in cooperative contexts, opinion changes and acceptance of decisions would be linked.

These findings have two key implications for public deliberation. First, the context of controversy influences group decisions; hence, the organizers of deliberations should consider discussion structures. In particular, they should consider how the structure of interaction during deliberation shapes the process and how goal interdependence affects outcomes. Second, when analyzing opinion changes through public deliberation, researchers should not limit their focus to whether the group-mean opinion is polarized or whether within-group opinions converge. They also need to examine group decisions and their evaluations in an integrated manner. Previous studies have often treated polarization and convergence as concerns to be avoided during deliberations [1,2,14,31]. Nevertheless, our findings suggest that, even when groups exhibit the same patterns of opinion change, the underlying processes may differ, highlighting the importance of assessing opinion change within a broader framework of group decision-making.

### Conclusion

This study conducted a group decision experiment that manipulated the context of controversy as a factor influencing opinion change and the evaluation of group decisions in public deliberation. These findings suggest that, even when competitive and cooperative contexts produce similar patterns of opinion change, the underlying processes may differ. Moreover, cooperative contexts led participants to evaluate group decisions more positively than competitive contexts. In conclusion, this study demonstrated that public deliberation can foster structured discussion.

### Supporting information

**S1 Table. All items and responses in the pre-questionnaire.**
(XLSX)

**S1 Appendix. Informed materials that participants received before the discussions.**
(PDF)

**S2 Table. All items and responses in the post-questionnaire.**
(XLSX)

**S3 Table. All results of the regression analysis in the Results section.**
(XLSX)

**S4 Table. Results of analysis regarding opinion change on soil reuse.**
(XLSX)

## Acknowledgments

We would like to thank Editage (www.editage.jp) for English language editing.

## Author contributions

**Conceptualization:** Yume Souma, Takashi Nakazawa, Tomoyuki Tatsumi, Yoshiko Arima, Susumu Ohnuma.

**Data curation:** Yume Souma.

**Formal analysis:** Yume Souma.

**Funding acquisition:** Yume Souma, Susumu Ohnuma.

**Investigation:** Yume Souma, Yukihide Shibata, Mie Tsujimoto, Qinglin Cui, Susumu Ohnuma.

**Methodology:** Yume Souma, Takashi Nakazawa, Tomoyuki Tatsumi, Yoshiko Arima, Susumu Ohnuma.

**Project administration:** Yume Souma, Susumu Ohnuma.

**Supervision:** Yume Souma, Takashi Nakazawa, Tomoyuki Tatsumi, Yoshiko Arima, Susumu Ohnuma.

**Validation:** Yume Souma.

**Visualization:** Yume Souma.

**Writing – original draft:** Yume Souma.

**Writing – review & editing:** Yukihide Shibata, Mie Tsujimoto, Qinglin Cui, Takashi Nakazawa, Tomoyuki Tatsumi, Yoshiko Arima, Susumu Ohnuma.

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
