## [Decision Letter · Decision Letter 0]

4 Feb 2026

Dear Dr. Souma,

Thank you for submitting your manuscript to PLOS ONE. After careful consideration, we feel that it has merit but does not fully meet PLOS ONE’s publication criteria as it currently stands. Therefore, we invite you to submit a revised version of the manuscript that addresses the points raised during the review process.

We look forward to receiving your revised manuscript.

Kind regards,

Sakae Kinase, Ph.D.

Academic Editor

PLOS One

**Journal Requirements:**

“This work was partially supported by the Graduate Grant Program of the Graduate 778 School of Humanities and Human Sciences, Hokkaido University, awarded to the first 779 author. “

“This work was supported by Environment Research and Technology Development Fund (https://www.env.go.jp/policy/kenkyu/suishin/english/gaiyou/index.html) (Grant Number [JPMEERF22S20907] and [JPMEERF20251001] to S.O.); the Japan Science and Technology Agency (https://www.jst.go.jp/EN/) (Grant Number [JPMJCR20D1] to S.O.); the Japan Society for the Promotion of Science (https://www.jsps.go.jp/english/) (Grant Number [23K22343] to S.O.; Grant Number [24KJ0296] to Ym.So.)

These funders had no role in study design, data collection and analysis, decision to publish, or preparation of the manuscript.”

5. We note that there is identifying data in the Supporting Information file <S1_Table.xlsx>. Due to the inclusion of these potentially identifying data, we have removed this file from your file inventory. Prior to sharing human research participant data, authors should consult with an ethics committee to ensure data are shared in accordance with participant consent and all applicable local laws.

-Location data

Please remove or anonymize all personal information, ensure that the data shared are in accordance with participant consent, and re-upload a fully anonymized data set. Please note that spreadsheet columns with personal information must be removed and not hidden as all hidden columns will appear in the published file.

**Additional Editor Comments:**

This manuscript has been carefully considered by two referees. The comments indicate that some fundamental revisions are necessary before the paper can again be considered for publication in PLOS One. Please carefully consider the comments. It is recommended that you revise the paper accordingly and re-submit in accordance with the Instructions to Authors.

Reviewers' comments:

Reviewer's Responses to Questions

**Comments to the Author**

1. Is the manuscript technically sound, and do the data support the conclusions?

Reviewer #1: Yes

Reviewer #2: Partly

2. Has the statistical analysis been performed appropriately and rigorously?

Reviewer #1: Yes

Reviewer #2: Yes

3. Have the authors made all data underlying the findings in their manuscript fully available?

Reviewer #1: Yes

Reviewer #2: Yes

4. Is the manuscript presented in an intelligible fashion and written in standard English?

Reviewer #1: Yes

Reviewer #2: Yes

Reviewer #1: Well written paper and very good description of the approach. Really useful for improving public decision-making in different domains.

Two points could be considered for additional discussion on the approach: 1) the discussion is focussed on final disposal while at the beginning the issue of reuse of soil is also mentioned. It should be useful to explain the focus of the study with regard to the broader issue. 2) As mentioned at the end in terms of limitation, the participants were "only" students. Although this is not representative of the citizens concerned with this issue, it could be interesting to highlight the usefulness to engage young generation in such a topic which has long-term challenges. In addition, the issue of solidarity between regions and accross generations could be discussed or mentioned.

More detailed comments: Fukushima is used sometimes in reference to the Prefecture, sometimes to the NPP, sometimes it is mentioned Fukuhsima Prefecture. An homogenisation of the term used would be helpful to understand where we are. On line 248, please change "limit" by "value". This is not a dose limit, but a radiological criteria.

Reviewer #2: The paper was written in standard English, but writing style could have been improved. Length of paper was much longer than what should be required to concisely present results of their study. Study design could also have been strengthened, assumptions and limitations more fully explored, and study would have benefited from some cross comparison to case studies that used competitive and collaborative approaches to decision-making, and/or a more fulsome discussion of how results compare to similar studies or approaches conducted by other scientists.

.

Reviewer #1: **Yes:** Thierry SCHNEIDERThierry SCHNEIDERThierry SCHNEIDERThierry SCHNEIDER

Reviewer #2: No

---

## [Author Response · Author response to Decision Letter 1]

16 Mar 2026

Thank you very much for your thoughtful review and constructive comments on our manuscript entitled “Effects of controversial contexts on opinion changes through discussion and evaluation of decisions: A group decision experiment regarding the issue of removed soil in Fukushima” (Manuscript ID: [PONE-D-25-63386]).

We have carefully considered the reviewers' feedback and revised the manuscript accordingly. For a detailed point-by-point response to each comment, please refer to the attached file "Response_to_Reviewers.docx".

---

## [Decision Letter · Decision Letter 1]

22 Mar 2026

Effects of controversial contexts on opinion changes through discussion and evaluation of decisions: A group decision experiment regarding the issue of removed soil in Fukushima

PONE-D-25-63386R1

Dear Dr. Souma,

We’re pleased to inform you that your manuscript has been judged scientifically suitable for publication and will be formally accepted for publication once it meets all outstanding technical requirements.

Kind regards,

Sakae Kinase, Ph.D.

Academic Editor

PLOS One

Additional Editor Comments (optional):

I have much pleasure in recommending this paper for publication. The manuscript has been substantially with changes according to reviewers' comments.

Reviewers' comments:

Reviewer's Responses to Questions

**Comments to the Author**

Reviewer #1: All comments have been addressed

2. Is the manuscript technically sound, and do the data support the conclusions?

Reviewer #1: Yes

3. Has the statistical analysis been performed appropriately and rigorously?

Reviewer #1: Yes

4. Have the authors made all data underlying the findings in their manuscript fully available?

Reviewer #1: Yes

5. Is the manuscript presented in an intelligible fashion and written in standard English?

Reviewer #1: Yes

Reviewer #1: Good reviesed paper. The authors have integrated the comments and the paper is now in a good format with clear explanation on the approach including a relevant discussion of the results.

.

Reviewer #1: **Yes:** SCHNEIDER ThierrySCHNEIDER ThierrySCHNEIDER ThierrySCHNEIDER Thierry

---

## [Editor Report · Acceptance letter]

PONE-D-25-63386R1

PLOS One

Dear Dr. Souma,

I'm pleased to inform you that your manuscript has been deemed suitable for publication in PLOS One. Congratulations! Your manuscript is now being handed over to our production team.

Kind regards,

on behalf of

Professor Sakae Kinase

Academic Editor

PLOS One